# Correlations between Ratings and Technical Measurements in Hand-Intensive Work

**DOI:** 10.3390/bioengineering10070867

**Published:** 2023-07-21

**Authors:** Gunilla Dahlgren, Per Liv, Fredrik Öhberg, Lisbeth Slunga Järvholm, Mikael Forsman, Börje Rehn

**Affiliations:** 1Section of Sustainable Health, Department of Public Health and Clinical Medicine, Umeå University, S-901 87 Umeå, Sweden; 2Radiation Physics, Department of Radiation Sciences, Umeå University, S-901 87 Umeå, Sweden; 3IMM Institute of Environmental Medicine, Karolinska Institutet, S-171 77 Stockholm, Sweden; 4Division of Ergonomics, School of Engineering Sciences in Chemistry, Biotechnology and Health, KTH Royal Institute of Technology, Hälsovägen 11C, S-141 57 Huddinge, Sweden; 5Department of Community Medicine and Rehabilitation, Physiotherapy, Umeå University, S-901 87 Umeå, Sweden

**Keywords:** ergonomics, humans, musculoskeletal disorders, upper extremity, wrist, hand, occupational health, risk, exposure, rating, psychophysics, hand intensity, inertial measurement units, electromyography

## Abstract

An accurate rating of hand activity and force is essential in risk assessment and for the effective prevention of work-related musculoskeletal disorders. However, it is unclear whether the subjective ratings of workers and observers correlate to corresponding objective technical measures of exposure. Fifty-nine workers were video recorded while performing a hand-intensive work task at their workplace. Self-ratings of hand activity level (HAL) and force (Borg CR10) using the Hand Activity Threshold Limit Value^®^ were assessed. Four ergonomist observers, in two pairs, also rated the hand activity and force level for each worker from video recordings. Wrist angular velocity was measured using inertial movement units. Muscle activity in the forearm muscles flexor carpi radialis (FCR) and extensor carpi radialis (ECR) was measured with electromyography root mean square values (RMS) and normalized to maximal voluntary electrical activation (MVE). Kendall’s tau-b correlations were statistically significant between self-rated hand activity and wrist angular velocity at the 10th, 50th, and 90th percentiles (0.26, 0.31, and 0.23) and for the ratings of observers (0.32, 0.41, and 0.34). Significant correlations for force measures were found only for observer-ratings in five of eight measures (FCR 50th percentile 0.29, time > 10%MVE 0.43, time > 30%MVE 0.44, time < 5% −0.47) and ECR (time > 30%MVE 0.26). The higher magnitude of correlation for observer-ratings suggests that they may be preferred to the self-ratings of workers. When possible, objective technical measures of wrist angular velocity and muscle activity should be preferred to subjective ratings when assessing risks of work-related musculoskeletal disorders.

## 1. Introduction

To develop effective injury prevention approaches for work-related musculoskeletal disorders (MSDs) from hand-intensive work, accurate quantitative risk assessments of hand activity and related force levels among workers are essential.

Exposure to hand-intensive work is common in the working population. For example, according to data from the US Bureau of Labor Statistics, Washington, DC, USA, approximately 30% of all workers are in occupational sectors that can be hand-intensive [1], and the percentage of hand-intensive occupations in Europe is similar [2]. Studies have shown that repetitive or constrained hand-intensive work increases the risk of MSDs of the neck, shoulder [3,4], hand, and arm [3,4,5]. 

Examples of sectors characterized by hand-intensive work and high incidence of MSDs in the upper extremities are slaughter [6], assembly, industrial production, processing, food processing industry, dentistry [4,7], and cleaning [4]. The specific types of MSDs may vary by sector and gender [8]. Carpal tunnel syndrome (CTS) is a particularly common diagnosis in workers performing hand-intensive work [5]. This syndrome is also recognized as an occupational disease in the European Union [9]. Exposure to repetitive hand movements [7,10,11,12,13,14] and high hand forces are described as risks for CTS [7,11,13,15]. Specifically, loads on the hand of more than 4 kg are associated with an increased risk for CTS [7]. Other risk factors are working with the wrist in flexion or extension, vibrating tools, and these combined exposures [7,11,14]. Among patients surgically treated for CTS in Sweden, almost 60% are manual workers [16].

For detecting MSD risk exposure in workplaces, risk assessment of physical exposure is statutory in accordance with work environment legislation in most countries worldwide. Risk assessment is the first essential step in the risk management process to protect and prevent workers from work-related MSDs. For hand-intensive work risk assessment, there are four categories of methods with different aims, purposes, contexts, levels of objectivity, precision, user knowledge, and costs. 

The first category is subjective assessments, which are self-reports [17], interviews, surveys, and checklists, such as the Washington State Ergonomics Tool [18]. These subjective assessments aim to identify risk and are simple to use, inexpensive, and applicable for various work situations, but can be imprecise [19]. The second category is observational methods [20]. This type of method is the most widely used in workplaces by ergonomists in the occupational health service. For hand-intensive work, the Hand Activity Level Threshold Limit Value^®^ (HA TLV^®^) is a commonly used risk assessment [21,22], and predictive for CTS [23]. The HA TLV^®^ assesses the construct’s hand activity level (HAL) [21] and hand force (Borg CR-10) [24] by workers rating their perceived exertion of the work on rate–ratio scales normalized to 0–10. The rating scales are recommended to be assessed by a professional observer and the worker. Other observational risk assessments also involve similar hand activity and force exposures [25,26,27]. These are assessed by the ergonomist, and some involve the workers providing self-ratings of exposure using a checklist of items. The third category is technical measures, also called direct measurements, which is a more objective category [28,29]. Measures in this category use technical, lightweight devices to quantify exposure. For example, electrogoniometry has been widely used for kinematic wrist velocity measures. Muscular activity measured with surface electromyography (EMG) attached to the body is also widely used in research [29,30]. These devices provide exposure information through a continuous sampling of data. However, for some devices, the measurements and data analyses are still complicated and require experienced and skilled practitioners. In the fourth category, digital human modeling (DHM) [31] can be used before the work or the workstation that actually exists to simulate and predict the workload [32].

High-quality measures of the construct’s hand activity level and hand force exposures in risk assessment can provide essential knowledge about MSD risks for hand-intensive occupations. If the exposure measures are questionable, this will affect the conclusions from the risk assessment. Ratings of hand activity and hand force (Borg CR-10) exposures are commonly used to assess risk in hand-intensive risk assessment. Therefore, it is essential to investigate the correlative strength between these rated constructs and technical measures designed to assess hand activity and force. Additionally, such an investigation should be conducted within the specific context and target population for which these measures are intended. To the best of our knowledge, the correlation between the subjective ratings of workers and observers for hand activity and hand force, as well as the correlations between those ratings and objective technical measures in regular hand-intensive work is unclear. Hence, this study aimed to investigate correlations between self- and observer-rated hand activity and between those ratings and wrist angular motions, as well as between self- and observer-rated hand force and between those ratings and muscular activity in hand-intensive work tasks.

## 2. Materials and Methods

### 2.1. Study Sites and Participants

This study is part of a larger project [33]. The study sites were eight companies, where the participants (workers) were assessed during a regular hand-intensive work task with different intensities regarding hand repetition and force to provide a variation of hand-intensive physical exposure. The occupational sectors represented by the participants were warehouse work, pharmaceutical production, industrial assembly work, postal service mail, postal sorting terminal work, postal sorting direct mail, food production, and laboratory work.

The workers were informed about the study through information meetings at each company. Written informed consent was obtained from all workers involved in the study. The eligibility criteria required the workers to be working with hand-intensive work and have no difficulties working with their arms [33]. Sixty-seven workers met the eligibility criteria and volunteered to participate. Eight workers were excluded (not hand-intensive work n = 2; simulated, mimicked hand-intensive work n = 2; illness n = 4). There were 59 workers (29 women and 30 men, respectively) included in the analyses. Work tasks are described in Table 1.

### 2.2. Experimental Protocol

The employers at each company were first contacted by phone, informed about the study, and asked permission to assess risk exposure in employees with hand-intensive work tasks in their regular work. Each employer signed a letter of intent. Next, the employer and workers participated in an informational meeting on the study at their company (GD). The employer and workers, with the support of GD, identified hand-repetitive work tasks executed cumulatively > 4 h per day. In total, 18 unique work tasks were identified, see Table 1. The workers were first asked to answer a composite questionnaire (Section 2.3). On another occasion, each worker had a physical examination for musculoskeletal disorders (Section 2.3). Within 0–7 days, the worker performed the hand-intensive work task and was simultaneously recorded with inertial measurement units (IMUs), surface electromyography (EMG), and a video camera (Section 2.4). Immediately afterwards, the worker rated hand activity and force (Section 2.7) of the recorded work task, and the equipment was removed. The physical examination, preparation before the work task, and rating of the work task took place in a quiet room at the company location.

### 2.3. Composite Questionnaire and Clinical Examination

The workers answered a composite questionnaire about working years and hours per day, illness, sick leave (Work Ability Index) [34], stress (time pressure and general stress according to the Quick Exposure Check) [29], and The International Physical Activity Questionnaire short form (IPAQ short) [35]. Next, anthropometrics (hand grip strength, forearm length, and finger abduction width) of each worker were recorded, and a standardized physical examination for MSDs, Health Examination under adverse Conditions, HECO [36], was performed.

### 2.4. Technical Measurements

Wrist angular velocity was measured using IMUs and hand force with EMG. These technical measures have previously been used as gold standards for their respective assessments [37].

#### 2.4.1. Movement Velocity and Muscular Activity

The DELSYS^®^ Trigno^®^ Wireless Biofeedback System Duo (Delsys Inc., Natick, MA, USA) was used. Within that system, the triaxial gyroscopes of two IMUs (Trigno Avanti^TM^ Sensors, 27 × 37 × 13 mm, 14 g) were used to measure the wrist angular velocity [38] of the most active hand during the work task. The sampling frequencies of the gyroscopes were 296 Hz for the hand and 222 Hz for the forearm. Each gyroscope had a dynamic range of ±250 degrees per second (dps).

Two surface EMG electrodes (Trigno Duo Sensors, 25 × 12 × 7 mm, 21 g) (Figure 1), each with two silver bars (99.9% silver) and an inter-electrode space of 10 mm and 5 × 1 mm contact dimensions (5 mm^2^), measured the muscular activity of the forearm muscles flexor carpi radialis (FCR) and extensor carpi radialis (ECR). The EMG data were sampled at 2000 Hz and bandpass filtered between 20 and 450 Hz in accordance with SENIAM recommendations [39]. The EMG signal was transformed to root mean square (RMS) by using an onboard function within the Trigno system. The used window length was 0.1 s.

The RMS of the work task duration was then computed and normalized to the individual maximal voluntary electrical activation (MVE). Hence, this RMS-value is the square root of the average power of the EMG signal during the task [40].

#### 2.4.2. IMU and EMG Sensor Location, Preparation, and Attachment

The most active hand during the work task was measured (right n = 58, left n = 1). One IMU was located on the dorsal hand (in the midline on the third metacarpal) and the other on the dorsal forearm (distally in the midline of the radius and ulna) as close as possible to the wrist joint but without contacting the IMUs in maximal wrist extension (Figure 1, middle).

The FCR EMG electrode-pair was located at the forearm, three to four fingerbreadths distal to the midpoint of a line connecting the medial epicondyle and the biceps tendon [41]. The ECR electrode-pair was located on the radial side of the forearm, approximately one-third of the distance of the forearm length from the elbow in a line from the lateral epicondyle [42,43]. The muscle bellies were located through palpation during voluntary wrist flexion (FCR) and extension (ECR). The surface of the sensors was cleaned with 70% isopropyl alcohol (Medic) and air dried. They were then dressed with double-sided self-adhesives (Delsys Adhesive Sensor Interface, Mini Sensor Adhesive Interface). The skin was shaved (if hairy), sandpapered, and wiped with 70% isopropyl alcohol until a light red skin color. The silver sensor bars (Figure 1) were applied parallel to the muscle fiber direction on the FCR and ECR muscle bellies [44] and the IMUs on the hand and forearm locations. The IMUs and EMG cables were fixated on the forearm with tape (Fixomull stretch). The IMUs were covered with 5 and 10-cm self-adherent wrap (Biltema, 3MCobanTMNL) on the forearm and hand. For some participant workers, the surface EMG electrodes were covered with self-adherent wrap. The IMU and EMG connections were then tested.

#### 2.4.3. EMG Normalization

To enable a comparison of EMG activity in the same muscles between the workers, EMG RMSs were normalized. Prior to the work task, the worker performed three maximal voluntary isometric contractions (MVIC), and the highest obtained MVE was used for normalization [45]. The MVIC was tested with the worker comfortably seated on a chair by the side of a table on which the forearm rested at elbow height on an anti-slip cloth. The forearm was semipronated, with a straight wrist and the hand outside the edge of the table. The worker resisted a perpendicular pressure centered on the third metacarpal by a dynamometer (MicroFET, Hoggan Scientific, LLC, round attachment 4 cm diameter). For FCR, the pressure was applied voluntarily in the hand. For ECR, the pressure was applied at the opposing center dorsally of the hand. The worker was verbally encouraged to resist force gradually to a maximum. Three five-second MVICs were performed, followed by five seconds of rest, for each muscle. All MVICs were instructed by the same researcher (GD).

#### 2.4.4. Data Processing and Analyses

The recorded IMU and EMG data were imported and processed in a custom-written software in MATLAB R2022B (Update 2 (9.13.0.2105380, 26 October 2022, The MathWorks Inc., Natick, MA, USA). The wrist angular velocity was computed for the 50th and 90th percentiles (from the time of the work task) according to the flex method of wrist flexion-extension velocity according to Manivasagam and Yang [38], i.e., only the velocity around the axis of flexion-extension was used, and the lower arm velocity was subtracted from the hand in the same way. Before the wrist angular velocity was calculated, the data from the gyroscope were low-pass-filtered using a fourth-order Butterworth filter with a cut-off frequency of 5 Hz followed by down-sampling to 20 Hz. The wrist velocity was then calculated at each instance in time using the equation below:$$v=\sqrt{(g\;hand-g\;forearm}$$

It has been shown that this way of measuring the velocity with IMUs correlates well with goniometer-measured velocity (Manivasagam and Yang) [38]. For the EMG data, the normalized 50th percentile and relative time in the work task with > 10%MVE, > 30%MVE, and < 5%MVE representing recovery [46] were computed.

### 2.5. Video Recording of the Work Task

The worker was instructed to perform the work task for 15 min. The work was video recorded with one camera (Panasonic 4K, HC—VXF1). This time period is often used for observing individual repetitive work tasks comprising a workday. The worker was video recorded so that the face (mimics), upper body, hands and arms, and whole body were captured dynamically during the task.

### 2.6. Workers’ Ratings of Hand Activity and Force

The HAL and force (Borg CR-10) scales were first introduced to the worker during a clinical examination for familiarization. The researcher (GD) explained the scales verbally and also provided a laminated text displaying the written scales. Each worker then test-rated the scales. Immediately after the work task, the researcher (GD) repeated both scales verbally and presented the written scales as in the test session. The researcher asked the following questions: “You will now assess how active you have been with the hand that we measured during the time of the measurement. Which number fits best for you?”, and “You will now assess the level of force in the measured hand during the time of the measurement. Which number fits best for you?”. The worker then rated the HAL and force (Borg CR-10) for the measured hand.

### 2.7. Observers’ Ratings of Hand Activity and Force

Four experienced ergonomists, who also were registered physiotherapists (two women and two men), were contacted by phone, informed about the study, and invited to participate as observers. They formed two mixed-gender pairs for the ratings. Each observer pair was first instructed and “calibrated” in a one-hour session of rating HAL and force (Borg CR-10) for the measured hand according to HA TLV^®^. On the next occasion, each observer-pair assessed video recordings (15 min each) of 28 work tasks. For assessment, the pairs were provided with the hand activity level scale and the Borg CR-10 force rating scale and were instructed to assess these two constructs jointly. The order of the videos was mixed, and they were free to choose the order within the set time.

### 2.8. Statistical Analysis

All data were imported and processed in SPSS Statistics for Windows (Released 2021. IBM SPSS Statistics for Windows, Version 28.0 (28.0.1.1(15)), Armonk, NY, USA: IBM Corp). Descriptive statistics were computed on anthropometric and questionnaire/sociodemographic data (mean, SD, median, interquartile range). Kendall’s tau-b correlations (tau) were calculated between hand activity ratings and wrist velocity at the 50th and 90th percentiles, and between force ratings and EMG at the 50th percentile and time during the work task > 30%MVE. All correlation analyses adopted a significance level of 5% (*p* < 0.05), and all tests were 2-sided.

## 3. Results

### 3.1. Description of the Workers

The 59 workers’ mean age was 35 years (SD 12). Workers reported pain or complaints > 3 (numerical rating scale 0–10) on days when pain or complaints occurred during the last 12 months [28] for the neck or shoulder *n* = 33 (55.9%, and for the elbow or hand *n* = 18 (30.5%). All workers were non-smokers (Table 2).

### 3.2. Self- and Observer-Ratings

The hand activity and force level ratings 0–10 were categorized into three categories. There were no self-ratings of force in the interval 7–10, which was the highest exposure category (Table 3).

### 3.3. Wrist Velocity and Muscular Activity

Table 4 displays the distribution of the IMU-measured wrist angular velocities in degrees per second (°/s) in three categories (low, median, and peak). The 50th percentile wrist velocity was 20.3°/s. Time spent during the work task is provided in four categories for MVE for FCR and ECR. The category < 5% represents recovery by < 5%MVE. The ECR muscle activity was higher than FCR [47].

### 3.4. Hand Activity and Wrist Angular Velocity

Positive and significant correlations were found between self- and observer-rated hand activity and wrist angular velocities for all measures, which is shown in Table 5.

As an example, the 50th percentile values between the self- and observer-rated hand activity, together with the corresponding wrist angular velocity, are illustrated in Figure 2. These correlations were significant but low (0.31 and 0.41, respectively).

### 3.5. Hand Force and Muscular Activity

None of the self-reported force levels were significantly correlated to the four MVE categories in FCR and ECR. For FCR, the correlations between observer-rated force and MVE were positive and significant for two categories (>10%MVE, >30%MVE) and negatively correlated for one category (<5%MVE). For ECR, the correlations between observer-rated force levels and MVE were positive and significant in one category (>30%MVE) (Table 6).

The 50th percentile values between the self- and observer-rated hand force with corresponding FCR EMG %MVE are illustrated in Figure 3. These correlations were considered little if any (0.00 and 0.29, respectively). The self-rated correlation was not statistically significant (*p* = 0.975). The observer-rated correlation was significant (*p* = 0.004)

The 50th percentile values between the self- and observer-rated hand force with corresponding ECR EMG %MVE are illustrated in Figure 4. Neither the self- nor the observer-rated correlations were significant. The tau values were little if any (−0.10 and 0.11, respectively).

### 3.6. Missing Values

There were no missing values for the IMUs. However, there were three missing values for observer-rated HAL. For HAL, the same set of workers was used (n = 56) for correlation analyses between self- and observer-ratings and between those ratings and technical measures. For the EMG data, there was a data loss due to abnormally high values for FCR (n = 5) and ECR (n = 2).

## 4. Discussion

The main aim of this study was to investigate the correlations between subjective ratings of hand activity and force and technical measures during hand-intensive work tasks. To our knowledge, this is the first study of workers who have no work-reducing MSDs that investigates correlations between self- and observer-rated hand activity and wrist angular velocity with IMUs and force ratings and muscular activity with EMG in regular hand-intensive work tasks.

The main contribution of our study is the positively significant correlations between self- and observer-rated HAL and wrist angular velocity. Further, force ratings and muscular activity level were significantly correlated only for observers (positively in four, and negatively correlated in one of eight measures). For all significant correlations, observer-ratings correlated more strongly (hand activity tau 0.32–0.41, force tau 0.26–0.44) than the self-ratings of the workers (hand activity tau 0.23–0.31) for both hand activity and hand force. To our knowledge, there is only one previous field study, by Buchholz et al. [47], that investigated correlations of subjective and technical assessments of hand activity and force during regular work tasks. We will therefore compare our results with the findings from that study.

### 4.1. Hand Activity and Technical Measures

Our significant correlations between self-rated hand activity and technical measures are similar to Buchholz et al. [47] but somewhat lower (tau 0.23–0.31, versus Pearson (r2) 0.38). Methodological differences are that Buchholz et al. used another scale than HAL, asking “How would you rate the pace at which you worked?” without wording for the anchors 0–10. Also, Buchholz et al. measured wrist angular velocity with electrogoniometers. As far as we know, there are no comparable studies that have used IMUs. However, it has been shown that the way we measured velocity with IMUs correlates well with goniometer-measured velocity (Manivasagam and Yang) [38].

### 4.2. Hand Force and EMG

Our non-significant correlation for self-rated hand force is in opposition to the significant correlation found by Buchholz et al. for the musculus flexor digitorum superficialis. The difference in findings may be partially due to differing methodologies. Buchholz et al. included four work tasks in comparison to our eighteen tasks. The exposure variation may therefore have been higher in our study.

### 4.3. Comparing Ratings and Technical Measures

It was surprising to find that only observer-rated force and EMG measures were significantly correlated. Likewise, of all significant correlations, the observer-ratings were better correlated to technical measures than the workers’ self-ratings.

Further, our significant Kendall’s tau b correlations ranged from 0.26 to 0.47. The interpretations of the cut-off values for the sizes of the correlations are 0.00 to 0.30, as little if any correlation; 0.30 to 0.50, as low positive (negative) correlation; 0.50 to 0.70, as moderate positive (negative) correlation; 0.70 to 0.90, as high positive (negative) correlation; and 0.90 to 1.0, as very high positive (negative) correlation [48]. Using the numerical value for the size of the correlation may however better enable comparisons across studies.

Investigating correlations between hand activity and force ratings with corresponding technical measures is challenging. It is difficult to know whether ratings and technical assessments measure the same modality. Rating might reflect other factors, such as a combination of the rater’s perception of the physical exposures, psychosocial factors, and normalization. For self and observer-ratings it has been found that the reliability of these assessments can be negatively affected by variations in different raters’ perceptions of hand activity and force [49,50].

When self-rating, the distractions of executing a specific work task can impact the perception of the physical exertion which might influence the rating. Further, the worker can become normalized to the work task. Psychosocial factors related to the context can be that the worker experiences expectations influenced by insecurity in reporting higher exposure values, due to fear of negative job consequences. In contrast, observers (such as the ergonomists in this study), who are trained, experienced, and rate multiple workers, might have provided a broader perspective on hand activity and force levels when rating. Also, the ergonomists are not normalized to the work task. This may partially explain why our observer ergonomists rated closer to the technical measures in our study.

The direct method of using technical measures to assess the physiological response regarding muscular activity and wrist angular velocity is not influenced by the rater’s perception of physical exertion in ratings. Still, technical measures are also influenced by subjective decisions, such as proper detection, sensor placement, calibration, signal processing and interpretation [39], and technical failure [47].

### 4.4. Strengths

As far as we know, our study is the first to measure self- and observer-rated effort levels of hand activity level (HAL) and hand force (Borg CR-10) according to HA TLV^®^ in a range of different regular hand-intensive work tasks in ordinary work at the workplace and correlate ratings and technical measures of motion and muscular activity. Accuracy of these exposures in the work risk management process is essential since hand activity and hand force are well-established risk factors for MSDs in workers performing hand-intensive tasks. We regard the range of the force ratings according to the Borg CR-10 scale as reasonable, with self-ratings from the low–middle range (0.5–6) and observer-ratings from the low to high range (range 0.5–9). Buchholz et al. demonstrated a similarly low–middle range for self-rated force in their study, despite a strategic effort prior to technical measurements to recruit 20 workers with low, middle, and high exposure based on ratings (Borg CR-10). The analysis of the technical measurements and post-ratings in the study by Buchholz et al. indicated that their workers only used low–middle force rating levels. Therefore, we argue that the self-ratings in our study may be reasonable and not due to regression to the mean [51]. The complete IMU samples (no data loss), and only a loss of EMG samples from 4 (n = 4) of 59 workers in our study, strengthens our results. Further strengths of our study include standardization by using the same assessors, distinct data collection routines, and the use of identical sets of equipment.

### 4.5. Limitations

We chose to investigate in regular and various work tasks the way exposure rating is normally assessed in workplaces. Also, the workers were free to work in a comfortable, self-chosen body position, sitting or standing. Grip force level has been demonstrated to be affected by different postures like sitting and standing [52,53]. This introduces variations that most likely influenced our results, in contrast to a controlled laboratory-based study for a single work task. We chose FCR and ECR for EMG. The choice of which muscle/s should represent hand force is a delicate matter when using EMG in a variety of regular work tasks as in this study, with the risk of some misleading estimations [19]. To compensate for this, before the actual data collection, we performed some pretrials, testing several muscles, which showed that wrist flexors and extensors tended to work as synergists. In the present study, EMG normalization was conducted by using MVIC in standardized positions for FCR and ECR before the work task, similarly to how others have done [47,54,55]. Normalization to an MVIC may not accurately reflect the level of muscle activation during dynamic work due to several reasons (such as the difficulty of the worker in performing the MVIC). However, for comparing muscular activity across various work tasks in our workers, the MVIC for %MVE was motivated in our study. Finally, we had four experienced ergonomists as observers. A higher number of observers would likely result in more reliable results. Still, four observers are similar to others [19]. The generalizability of these results may be to similar work tasks, contexts, ratings, and technical measures. However, interpretation to other contexts and other hand activities and force exposures than in this study, such as in sectors of, e.g., assistant nurses or construction builders, is unknown and should be investigated.

### 4.6. Practical Applications and Implications

High-precision risk assessments of hand-intensive work may help to better identify risk exposures for MSDs. They may also facilitate our understanding of related injury mechanisms, enable assessment of changes over time, and help employers and employees to make informed decisions about how to control and manage physical work exposure risks. If a risk assessment underestimates risks, this could lead to a false sense of security and inadequate safety measures. This could in turn result in workers being exposed to preventable physical exposures leading to MSD complaints, diagnoses, disabilities, sick days, and, in the worst case, inability to work [56]. On the other hand, overestimation of risk can lead to unnecessary actions for risk elimination or risk reduction.

Choosing the relevant hand-intensive risk assessment method with its exposure items is a delicate matter for researchers and ergonomists. First, the purpose and context need to be considered. As mentioned in the introduction, if the purpose is to identify risks for CTS by rating hand activity and force, HA TLV^®^ is a valuable observational method because it has been shown to predict CTS [23]. This type of observational method can also be practical for large populations since it is quick, easy to use, and has a low cost. However, if the purpose of the risk assessment is to compare the risk for work-related MSDs to scientifically based action levels [46] for different groups (e.g., in sex or gender, between work tasks [4,30]) based on the physiological entities of the exposure (wrist angular velocity, %MVE), technical measures should be preferred. Then, choosing the relevant hand-intensive risk assessment method also requires knowledge of the exposures that are measured and how. In our case, it is not clear whether the workers’ and ergonomists’ ratings of hand activity and force capture the same modality as the measured wrist angular velocity and EMG. The modest correlations indicate that they measure somewhat different modalities. It is possible that subjective assessments of the rated perceived exertion of hand activity and hand force are not strictly limited to actual activity and force exposures. Instead, these might be influenced by their combinations and other psychosocial factors.

The modest correlations between ratings and technical measures in our study could suggest that the latter are to be preferred for exposure and risk assessments when available. Our study showed that all observer-ratings were better correlated to technical measures compared to the self-ratings of workers. Hence, using experienced ergonomists is preferable. This is particularly the case for force rating, which was significant only for observer-rated force.

In general, technical measurements are considered the gold standard for risk assessment compared to self-reports and observation methods. Technical measures, for their part, should, to a lesser extent, be influenced by human raters. These measures represent the physiological entities of wrist angular velocity and muscular activity. The linear outcome (degrees/second and %MVE) enables higher resolution compared to rating 11 levels. Hence, technical measures may be useful for practitioners to measure changes with higher resolution in exposure pre- and post-work risk-reducing interventions. The type of technical measures used in our study requires knowledge, costly equipment, and time for data collection and analysis, hence perhaps more useful in research or special cases. More user-friendly and cheaper IMUs have, however, been developed recently. Still, there is a need for corresponding technical equipment for muscular activity to meet the needs and requirements contextualized to occupational health clinicians.

In our study, the observer video rating was only used to assess hand activity and force. We had permission for filming from the employer and worker and did not experience difficulties filming the workers in their work tasks. Our general experience, however, is that video recording of certain sensitive work tasks is not allowed and that workers sometimes do not want to be filmed. Videos can be useful for the observer by giving them more time to decide on their rating. Also, it can increase the likelihood of detecting other risk factors, such as behaviors when executing work tasks, body postures, and possible influences of the working environment on the worker.

Exposure assessments are the fundamental elements of risk assessment. In ISO 31000:2018 [57] and ISO 45001:2018 [58], risk analysis corresponds to risk assessment as used in our study. According to these ISO standards, risk identification and risk analysis are the first crucial steps in the work risk management process and prevention at workplaces. Mere measurement alone is unlikely to bring about the desired reduction in MSD work risk and MSD prevention. When identified work risks and analyses are shared with the stakeholders (employer, workers, and others), the stakeholders can support a common understanding and shared values regarding work risk objectives and outcomes in their integrated contextualized risk management process. This involves a dynamic integrated iterative process reasoning around ´what will make this happen?’ and ‘how will this happen?’, specified in terms of What?, When?, Where?, How often?, and Involving Whom? to support the change process [59]. Hence, the initial risk analysis (or risk assessment) may also play a crucial role in follow-up, which helps to clarify whether the risk has been changed in terms of elimination or reduction, and to what extent. Therefore, risk assessment is an essential part of the risk management process to support the prevention of work-related MSDs.

## 5. Conclusions

Despite similarities between hand activity and hand force ratings and corresponding technical measures, their modest correlations suggest that they represent somewhat different modalities. In our study, the ratings of observers were more strongly correlated than self-ratings to technically assessed wrist motions. Additionally, only the ratings of observers were correlated to muscular activity. This suggests that if ratings are used in risk assessments of hand-intensive work, ratings of observers should be preferred to the ratings of the workers themselves. When possible, objective technical measures of wrist angular velocity and muscle activity should be preferred to subjective ratings when assessing risks of work-related musculoskeletal disorders. The understanding of rated and technically measured exposure and their correlation is crucial for clinicians as the first important step in the risk management process for work-related MSD prevention.

## Figures and Tables

**Figure 1 bioengineering-10-00867-f001:**
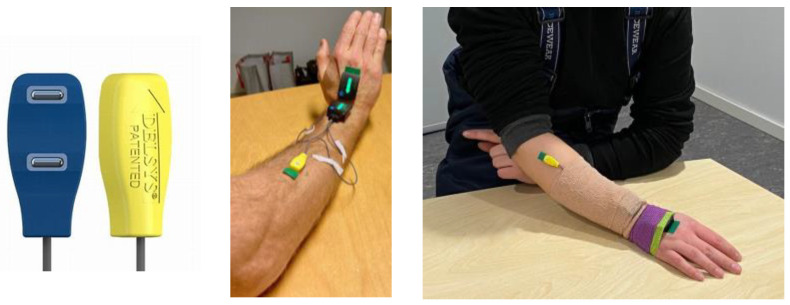
(**Left**): EMG electrodes; (**middle**) and (**right**): the attached IMU on the hand and distal forearm, and EMG electrodes on ECR (the FCR location is not visible) before and after being secured with tape and wrap.

**Figure 2 bioengineering-10-00867-f002:**
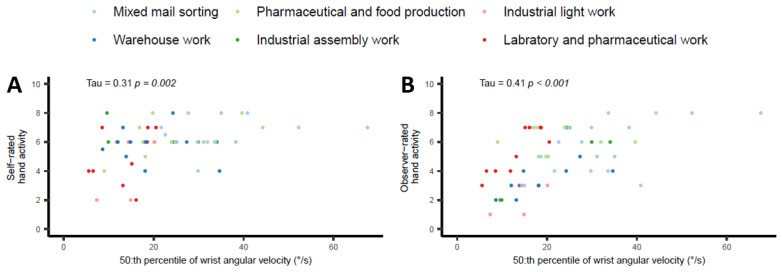
Hand activity self-ratings (**A**) and observer-ratings (**B**) for the 50th percentiles. The colors represent six occupational work task production groupings.

**Figure 3 bioengineering-10-00867-f003:**
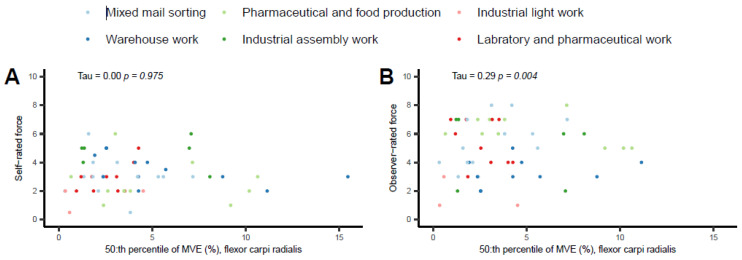
Hand force self-ratings (**A**) and observer-ratings (**B**) for the FCR 50th percentiles. The colors represent six occupational work task production groupings.

**Figure 4 bioengineering-10-00867-f004:**
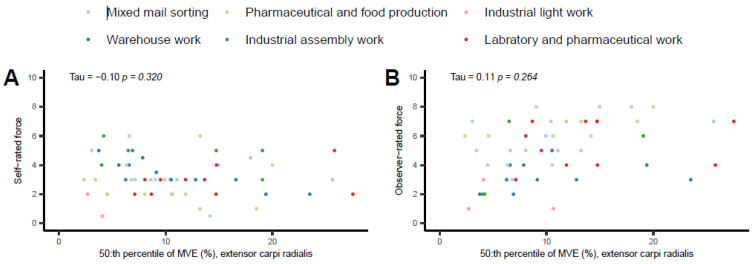
Hand force self-ratings (**A**) and observer-ratings (**B**) for the 50th percentiles of ECR EMG %MVE. The colors represent six occupational work task production groupings.

**Table 1 bioengineering-10-00867-t001:** The hand-intensive work tasks categorized into groups of occupational work tasks of production types. Each worker performed one task.

Grouped Work Tasks	Workers per Task (n = 59)
1. Mixed mail sorting	
Manual sorting of mail	9
Manual sorting of direct mail	4
Manual sorting of catalogues	4
2. Warehouse work	
Ranking of goods	5
Picking, base products, heavier load	4
Picking, fruit, vegetables, lighter load	3
3. Pharmaceutical and food production	
Manual decontamination of bags	4
Fluid inspections of bottles	2
Steamplicity, manual packaging of food portions	4
4. Laboratory and pharmaceutical work	
Cassette filling	4
Inspection, labelling, packaging of ampoules	2
Water filtration	2
Manual pipetting	2
5. Industrial assembly work	
Hose coupling	2
Hose winding	2
Wheeling	2
6. Industrial light work	
Paternoster picking	2
Small parts picking, scanning	2

**Table 2 bioengineering-10-00867-t002:** Description of the workers who participated in the study, *n* = 59.

	Workers
Demography, anthropometrics, and lifestyle	*n* = 59
Dominant hand	Right *n* = 55 (93.2%)
	Left *n* = 4 (6.8%)
Body weight, kg	82.9 (19.2) ^1^
Body height, cm	176.2 (10.6) ^1^
BMI	25.5 [23.1, 29.1] ^2^
Diagnoses from the neck and shoulder [29,30]	*n* = 13 (22.3%)
Tension neck syndrome *n* = 2, cervicalgia *n* = 1, thoracic outletsyndrome *n* = 1, acromioclavicular syndrome *n* = 4, biceps tendinitis *n* = 4 and supraspinatus tendinitis *n* = 1	
Diagnoses from the hand and arm [29,30]	*n* = 7 (11.9%)
De Quervain *n* = 2, overused hand syndrome *n* = 1, pronator teres syndrome *n* = 1, carpal tunnel syndrome *n* = 2 and ulnar nerve entrapment elbow *n* = 1	
Work exposure	
How many years of working experience do you have with hand-intensive tasks?	6.8 [2.0, 15.5] ^2^
How many hours per day do you work during a normal day with hand-intensive tasks, repeated movements and exertions?	5.4 (1.9) ^1^
How many hours per day do you work during an intensive day with hand-intensive tasks, repeated movements, and exertions?	7.0 [6.0, 8.0] ^2^

^1^ Mean (SD), ^2^ median, interquartile range.

**Table 3 bioengineering-10-00867-t003:** Self- and observer-rated hand activity and force in the 18 work tasks with numbers (*n*) represented in three categories (0–3, 4–6, and 7–10) of hand activity and force.

Variables	Min	Max	Mean (SD)	Ratings 0–3, *n*	Ratings 4–6, *n*	Ratings 7–10, *n*
Hand activity self-rated	1	8	5.9 (1.5)	4	37	18
Force self-rated	0.5	6	3.2 (1.3)	39	20	0
Hand activity observer-rated	1	8	5.0 (1.9)	15	25	16
Force observer-rated	0.5	9	3.5 (2.3)	29	22	5

**Table 4 bioengineering-10-00867-t004:** Wrist angular velocity in degrees per second and muscular activity as time spent in four categories of %MVE.

Variables	Median [Interquartile Range] ^1^, Mean (SD) ^2^
Wrist angular velocity	*°/s*
10th percentile	2.8 [2.0, 3.5] ^1^
50th percentile	20.3 [14.8, 29.9] ^1^
90th percentile	81.3 [67.3, 108.5] ^1^
Muscular activity, RMS	% time spent
Median %MVE FCR	3.1 [1.8, 5.4] ^1^
Time >10%MVE FCR	11.96 [4.5, 26.0] ^1^
Time >30%MVE FCR	0.8 [0.1, 4.0] ^1^
Recovery, proportion of time <5%MVE FCR	64.1 (22.6) ^2^
Median %MVE ECR	9.9 [6.5, 14.7] ^1^
Time >10%MVE ECR	47.6 [31.1, 67.1] ^1^
Time >30%MVE ECR	6.2 [2.2, 15.7] ^1^
Recovery, proportion of time <5%MVE ECR	29.4% (19.4) ^2^

**Table 5 bioengineering-10-00867-t005:** Kendall’s tau-b correlation and *p*-values (2-tailed) between self-ratings, observer-ratings, and the wrist angular velocity in the 10th, 50th, and 90th percentile (*n* = 56).

	10th	50th	90th
Self-rated hand activity	0.26	0.31	0.23
*p*-value	0.005	0.002	0.024
Observer-rated hand activity	0.32	0.41	0.34
*p*-value	<0.001	<0.001	<0.001

**Table 6 bioengineering-10-00867-t006:** Kendall’s tau-b correlations between force ratings and levels of MVE (*n* = 59). Bold indicates significant values.

		FCR				ECR		
	50th	>10%MVE	>30%MVE	<5%MVE	50th	>10%MVE	>30%MVE	<5%MVE
Self-rated hand force	0.00	0.05	0.09	0.005	–0.10	–0.08	0.02	0.01
*p*-value	0.975	0.650	0.388	0.963	0.320	0.419	0.854	0.963
Observer-rated hand force	0.29	0.43	0.44	–0.47	0.11	0.11	0.26	–0.09
*p*-value	0.004	<0.001	<0.001	<0.001	0.264	0.267	0.010	<0.537

## Data Availability

Not applicable.

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
