# Peer review of "Correlations between Ratings and Technical Measurements in Hand-Intensive Work"

_bioengineering, 2023, doi:10.3390/bioengineering10070867_

Round 1

Reviewer 1 Report

The study entitles “Correlations Between Ratings and Technical Measurements in Hand-Intensive Work “ by Dahlgren et al. has an interesting aim and could be a rational for developing effective injury prevention approaches for work-related musculoskeletal disorders from hand-intensive work.

Please find my critiques and suggestions:

It is mentioned that the study doesn’t aim to compare the risk for work-related MSDs to scientifically based action levels between different groups e.g., sex and gender. However, considering this variable while plotting the results will serve the purpose of the study better. It will be interesting to see if the data from male and female participants show similar or differential patterns of responses. In my opinion in order to develop an effective injury prevention approach and promote occupational health, it is very important to consider sex as one of the parameters of variation.

Therefore, it is suggested that authors should include these comparisons in the result section and highlight any significant findings in the discussion. Or should state it clearly if there have been no significant changes or similarities found between male and female participants.

Reviewer 2 Report

The authors investigate the correlations between subjective, objective ratings of hand activity, and force and technical measures during work tasks. They found only limited correlations between ratings and technical measures.

Evaluating (both objectively and subjectively) hand activity is very challenging and the use of the described technique may be clinically helpful.

1) In the discussion it would be helpful to add more detail to the ways this technique could be applied including the practical disadvantages -such as obtaining patient’s video etc..

2) The figures are slightly confusing and could be partially omitted.

Reviewer 3 Report

Dear authors,

I have two main suggestions to make your manuscript better:

1. In my view, the key aim of your study has been to prevent occupational diseases (MSDs, in particular). I propose to highlight this aim in the introduction (lines 98 to 101). In addition, it is worth highlighting in the conclusions how your study contributed to achieving the objective.

2. Your study will be easier for OSH practitioners to understand by referencing the two core standards - ISO 45001: 2018 and ISO 31000: 2018. Both standards constitute the normative basis of practical work in industry.

Minor editing of English language required. Please polish terminology according to ISO 45001 standard.

Round 2

Reviewer 1 Report

The comments have addressed well.